# A New Light on Potential Therapeutic Targets for Colorectal Cancer Treatment

**DOI:** 10.3390/biomedicines9101438

**Published:** 2021-10-10

**Authors:** Wei-Lun Tsai, Chih-Yang Wang, Yu-Cheng Lee, Wan-Chun Tang, Gangga Anuraga, Hoang Dang Khoa Ta, Yung-Fu Wu, Kuen-Haur Lee

**Affiliations:** 1PhD Program for Cancer Molecular Biology and Drug Discovery, College of Medical Science and Technology, Taipei Medical University and Academia Sinica, Taipei 11031, Taiwan; mebar1995@gmail.com (W.-L.T.); g.anuraga@unipasby.ac.id (G.A.); d621109004@tmu.edu.tw (H.D.K.T.); 2PhD Program for Cancer Molecular Biology and Drug Discovery, College of Medical Science and Technology, Taipei Medical University, Taipei 11031, Taiwan; chihyang@tmu.edu.tw (C.-Y.W.); yeas0310@hotmail.com (W.-C.T.); 3Graduate Institute of Cancer Biology and Drug Discovery, College of Medical Science and Technology, Taipei Medical University, Taipei 11031, Taiwan; 4Graduate Institute of Medical Sciences, College of Medicine, Taipei Medical University, Taipei 11031, Taiwan; yclee0212@tmu.edu.tw; 5Department of Statistics, Faculty of Science and Technology, Universitas PGRI Adi Buana, Surabaya 60234, East Java, Indonesia; 6National Defense Medical Center, Department of Medical Research, School of Medicine, Tri-Service General Hospital, Taipei 11490, Taiwan; qrince@yahoo.com.tw; 7Cancer Center, Wan Fang Hospital, Taipei Medical University, Taipei 11031, Taiwan

**Keywords:** colorectal cancer, CRNDE, MiR-29b-3p, ANGPTL4, autophagy, lipid metabolism

## Abstract

The development and progression of colorectal cancer (CRC) involve changes in genetic and epigenetic levels of oncogenes and/or tumor suppressors. In spite of advances in understanding of the molecular mechanisms involved in CRC, the overall survival rate of CRC still remains relatively low. Thus, more research is needed to discover and investigate effective biomarkers and targets for diagnosing and treating CRC. The roles of long non-coding RNAs (lncRNAs) participating in various aspects of cell biology have been investigated and potentially contribute to tumor development. Our recent study also showed that CRNDE was among the top 20 upregulated genes in CRC clinical tissues compared to normal colorectal tissues by analyzing a Gene Expression Omnibus (GEO) dataset (GSE21815). Although CRNDE is widely reported to be associated with different types of cancer, most studies of CRNDE were limited to examining regulation of its transcription levels, and in-depth mechanistic research is lacking. In the present study, CRNDE was found to be significantly upregulated in CRC patients at an advanced TNM stage, and its high expression was correlated with poor outcomes of CRC patients. In addition, we found that knocking down CRNDE could reduce lipid accumulation through the miR-29b-3p/ANGPTL4 axis and consequently induce autophagy of CRC cells.

## 1. Introduction

Cancers are leading causes of death globally, with an estimated 9.6 million deaths in 2018 [1]. One of the frequently encountered malignant tumors is colorectal cancer (CRC) with the third highest incidence of cancer worldwide [2]. The development and progression of CRC involve changes in genetic and epigenetic levels of oncogenes and/or tumor suppressors [3]. In spite of advances in treatments and our understanding of the molecular mechanisms involved in CRC, overall survival (OS) rates of CRC patients still remain relatively low [4]. Since mechanisms of the progression of CRC are not fully understood, more research is needed to discover and investigate effective biomarkers and targets for diagnosing and treating CRC.

High-throughput genome-scale studies demonstrated that more than 90% of DNA sequences in the human genome are actively transcribed [5]. However, only 1% of the human genome is composed of protein-coding genes [6]. This indicates that about 70%–90% of the genome can be transcribed at some point during development to produce a large transcriptome of non-coding (nc)RNAs, and 4%–9% of these are transcribed to yield many short or long RNAs with limited protein-coding capacities [7]. Long non-coding (lnc)RNAs with transcripts of >200 nucleotides and which lack an open reading frame are defined as lncRNAs [6,8]. Many studies have reported that lncRNAs participate in various aspects of cell biology and potentially contribute to tumor development [9]. Dysregulation of lncRNAs commonly exerts impacts on cellular functions such as cell proliferation, resistance to apoptosis, induction of angiogenesis, promotion of metastasis, and evasion of tumor suppressors [9,10]. Colorectal neoplasia differentially expressed (CRNDE) was originally discovered as an upregulated lncRNA in CRC, while in contrast, it shows little to no expression in the normal colon epithelium [11]. Our recent study also showed that *CRNDE* was among the top 20 upregulated genes in CRC clinical tissues compared to normal colorectal tissues according to an analysis of a Gene Expression Omnibus (GEO) dataset (GSE21815) (unpublished data from [12]). Moreover, an increasing number of studies suggested that CRNDE might be a potential diagnostic biomarker and prognostic predictor due to its high sensitivity and specificity in cancer tissues, and its upregulation was significantly correlated with larger tumor sizes, lymph node metastasis, distant metastasis, and worse OS [13,14,15,16]. Although CRNDE was widely reported to be associated with different types of cancer, most studies on CRNDE only investigated regulation of its transcription levels, and in-depth mechanistic research is lacking [17]. Thus, in the present study, we aimed to investigate the detailed mechanisms of CRNDE in CRC tumorigenesis.

Autophagy plays a crucial role in providing a mechanism for recycling proteins, lipids, and organelles during cellular conditions of stress and starvation. In the development of human diseases, autophagy was shown to be a double-edged sword. In cancer cells, oncogenes and severe stress conditions drive profound upregulation of autophagy to temporarily promote cell survival [18]. Conversely, if cellular stress leads to continuous or excessively induced autophagy, cell death will ensue [19]. In addition, an elevated level of autophagy was observed in many cancer cells under stressed conditions, suggesting that autophagy may have a cytoprotective role and function as a oncogenic mechanism in certain tumor development stages [20]. However, little is known about the biological function and significance of the potential molecular mechanism of the role of CRNDE in autophagy in CRC.

In the current study, in order to investigate the potential role of CRNDE in regulating autophagy, we first investigated the role of CRNDE in CRC cells, and consequently, characterized loss of CRNDE-triggered autophagy through regulation of metabolism signaling. Importantly, we found that knocking down CRNDE could reduce lipid accumulation through the miR-29b-3p/ANGPTL4 axis and consequently induce autophagy of CRC cells. Our study may provide new clues on molecular events among CRNDE, miR-29b-3p, and ANGPTL4, thereby shedding new light on potential therapeutic targets for CRC treatment.

## 2. Materials and Methods

### 2.1. Chemicals, Reagents, and Antibodies

Methanol, crystal violet, and chloroquine (CQ) were obtained from Sigma-Aldrich (St. Louis, MO, USA). Rabbit polyclonal antibodies against cyclin-dependent kinase 4 (CDK4), poly(ADP-ribose) polymerase (PARP), Bcl-2, Bid, Thr(P)172-adenosine monophosphate-activated protein kinase (AMPK), AMPK, Ser(P)2448-mammalian target of rapamycin (mTOR), mTOR, Ser(P)79-acetyl-CoA carboxylase (ACC), ACC, fatty acid synthase (FAS), microtubule-associated light chain 3 (LC3), and p62 were obtained from Cell Signaling (Beverly, MA, USA); p21 and cyclin D1 were from Santa Cruz Biotechnology (Santa Cruz, CA, USA); Ser(P)872-hydroxymethylglutaryl-CoA reductase (HMGCR), and HMGCR were from Millipore (Billerica, MA, USA); cyclin D1 was from Santa Cruz Biotechnology; and ANGPTL4 was from Abcam (Cambridge, MA, USA). Mouse monoclonal antibodies against beta-actin and caspase-3 were, respectively, purchased from MP Biomedicals (Irvine, CA, USA) and Imgenex (San Diego, CA, USA).

### 2.2. Cell Culture

CRC cell lines were provided by the Graduate Institute of Cancer Biology and Drug Discovery, Taipei Medical University. All CRC cell lines were cultured in RPMI-1640, supplemented with 10% fetal bovine serum (FBS) and antibiotics (all from Thermo Fisher Scientific, Waltham, MA, USA), and were maintained at 37 °C in a humidified atmosphere containing 5% CO_2_.

### 2.3. Cell Transfections

Two individual CRNDE (CRNDE 1 and 2) and scrambled negative control small interfering (si)RNAs were purchased from Invitrogen (Carlsbad, CA, USA);the green fluorescent protein (GFP)-CRNDE plasmid was from Genscript Biotech (Piscataway, NJ, USA);and has-miR-29b-3p miScriptmiRNAMimics was from Qiagen (Valencia, CA, USA), and was transfected into cells using the jetPRIME transfection reagent (Polyplus-transfection, New York, NY, USA) according to the manufacturer’s instructions. Sequences of the siRNAs are described in Appendix A.

### 2.4. Cell Viability Assay

Cell viability was determined with the crystal violet-staining method, as described previously [21]. In brief, the oligonucleotide (100 nM) was introduced into 5 ×10^5^ dissociated cells by the jetPRIME transfection reagent according to the manufacturer’s instructions. Next, cells were plated in 96-well plates at 3000 cells/mL after transfection with control siRNA or siCRNDE for 48 h. After cells had grown for 48 h, cells were stained with 0.5% crystal violet for 10 min at room temperature. Next, the plates were washed with tap water three times. After drying, cells were lysed with a 0.1 M sodium citrate solution (Sigma-Aldrich, St. Louis, MO, USA), and the absorbance was measured at 550 nm on a microplate reader.

### 2.5. Focal Formation Assays

HCT-116 cells were seeded at 4000 cells/well in six-well dishes and grown overnight after transfection with control siRNA or siCRNDE for 48 h. The medium was changed every 3 days. After 11 days, cells were fixed and stained with 0.5% crystal violet. Foci of >5 mm in size were counted, and average focal counts and standard deviations (SDs) were calculated.

### 2.6. Cell Cycle Analysis

Cells were transfected with control siRNA or siCRNDE for 48 h, and a cell-cycle analysis was performed. Harvested cells were washed in phosphate-buffered saline (PBS), and 200 µL of Muse cell cycle reagent (EMD Millipore, Billerica, MA, USA) was added. Cells were incubated for 30 min at room temperature in the dark. The cell cycle distribution was analyzed by a Muse Cell Analyzer (EMD Millipore).

### 2.7. Apoptosis Assay

An apoptosis assay was conducted using a flow cytometry-based approach. In order to evaluate the effect of siCRNDE in inducing apoptosis, HCT116 cells (2.5 × 10^5^) were transfected with siCRNDE for 48 h, and then cells were collected in culture medium, mixed with the Muse Annexin V and Dead Cell Reagent, and analyzed with a Muse Cell Analyzer (EMD Millipore).

### 2.8. Autophagy Cytofluorimetric Analysis

To examine autophagic flux, we used a Muse™ Red Fluorescent Protein (RFP)-LC3 Reporter Autophagy Assay Kit, which contained the stably expressing RFP-LC3 Reporter U2OS cell line (EMD Millipore) to measure and track LC3 levels within cells after transfection with siCRNDE according to the manufacturer’s instructions. The analysis was performed using a Muse Cell Analyzer (EMD Millipore).

### 2.9. Glucose Uptake Detection

Cells were transfected with control siRNA or siCRNDE for 48 h. After that, glucose uptake was assessed using a glucose uptake assay kit (Abcam, Cambridge, UK) following the manufacturer’s instructions. Briefly, cells were starved in serum-free medium overnight and then placed in Krebs-Ringer-Phosphate-HEPES buffer with 2% bovine serum albumin (BSA) for 20 min. Next, the glucose analog 2-deoxyglucose (2-DG) was added to cells, and the accumulated 2-DG6P was oxidized to generate NADPH, which resulted in oxidation of the substrate. The oxidized substrate could then be detected at an OD of 412 nm.

### 2.10. Glycolysis Stress Test

The extracellular acidification rate (ECAR) for assessing cell glycolysis or the glycolytic capacity was determined using a Seahorse XF Glycolysis Stress Test Kit (Agilent, Santa Clara, CA, USA) according to the manufacturer’s instructions. Cells were transfected with control siRNA or siCRNDE for 48 h, trypsinized, and seeded into Seahorse XF cell culture plates. The ECAR was detected in an XF96 Analyzer (Agilent).

### 2.11. BODIPY Staining

Cells were transfected with control siRNA or siCRNDE for 48 h. After that, cells were fixed in 3.7% paraformaldehyde for 60 min. Next, cells were incubated with 4,4-Difluoro-1,3,5,7,8-Pentamethyl-4-Bora-3a,4a-Diaza-*s*-Indacene (BODIPY^505/515^) (1 μg/mL, ThermoFisher Scientific, Waltham, MA, USA) to measure lipid accumulation (60 min at room temperature). Images were examined on a microscope (Olympus, Tokyo, Japan). To analyze the percentage of lipid droplet-containing cells, numbers of cells with BODIPY^+^ lipid droplets were counted, and the percentage of BODIPY^+^ cells was calculated.

### 2.12. Quantitative Reverse-Transcription Polymerase Chain Reaction (RT-qPCR)

Total RNA was extracted from cell lines with or without treatment with a Qiagen RNeasy kit and Qiashredder columns according to the manufacturer’s instructions (Qiagen, Valencia, CA, USA). One microgram of total RNA was reverse-transcribed to complementary (c)DNA using a Reaction Ready™ First Strand cDNA Synthesis Kit (SABiosciences, Frederick, MD, USA). An RT-qPCR was carried out in an Applied Biosystems StepOnePlus™ Real-Time PCR System (Applied Biosystems, Foster City, CA, USA) using an automated baseline and threshold cycle (Ct) detection. Sequences of specific primers for each gene are listed in Appendix A. To detect expression levels of miR-134-5p and miR-29b-3p, specific products were amplified and detected with the cycle profile according to the miScript PCR starter kit (Qiagen). The relative gene expression level was calculated by comparing the cycle times for each target PCR. Target PCR Ct values were normalized by subtracting the U6 ribosomal (r)RNA Ct value, which was the internal control.

### 2.13. Luciferase Reporter Assays

To investigate the relationship between CRNDE and miR-29b-3p, the wild-type (WT) CRNDE or mutant (Mut) CRNDE that had the predicted miR-29b-3p-binding site was established and integrated into a pMIR-luciferase reporter vector to form the pMIR-CRNDE-wild type (CRNDE-WT) or pMIR-CRNDE-mutant (CRNDE-Mut) reporter vector. Luciferase reporter assays were performed with the Luciferase Reporter Assay System (Promega, Madison, WI, USA). The CRNDE-WT or CRNDE-Mut vector was co-transfected into HCT-116 cells with miR-29b-3p mimics or a negative control using the jetPRIME transfection reagent. Subsequent to transfection for a period of 48 h, luciferase activities were measured in accordance with the manufacturer’s guidelines.

### 2.14. Western Blotting

Cell lines were placed in lysis buffer for 1 h at 4 °C. Protein samples were analyzed using different percentages of sodium dodecylsulfate (SDS)-polyacrylamide gel electrophoresis (PAGE), as previously described [22].

### 2.15. Gene Set Enrichment Analysis (GSEA)

We used GSEA vers. 4.0 to perform a GSEA on public resources (GSE89985) to investigate gene set enrichment, which was related to CRNDE-knockdown (KD) in DLD-1 CRC cells. Gene sets were obtained either from the molecular signatures database (MSigDB) hallmark gene set [23] or from a public resource (GSE89985).

### 2.16. In Situ Hybridization (ISH) and Immunohistochemistry (IHC)

ISH and the IHC assay used a colon adenocarcinoma tissue array (CO1505) purchased from US Biomax (Rockville, MD, USA), which contained 50 cases of CRC tissues with matched adjacent tissues as the controls. The ISH assay was performed as previously described [24]. In brief, the colon tissue array was fixed for 24 h in 4% paraformaldehyde. The expression of CRNDE or miR-29b-3p was detected using a digoxigenin (Dig)-conjugated CRNDE or miR-29b-3p probe (Toson Technology, Hsinchu, Taiwan) on paraffin-embedded colon tissues, which had the probe sequences of 5′-CCTCAGTTGTCACGCAGAAG-3′ for CRNDE [25] and 5′-CACTGATTTCAAATGGTGCTA-3′ for miR-29b-3p. The ISH assay was performed as described previously [26]. In brief, human colorectalspecimens were fixed in 4% paraformaldehyde for 24 h. CRNDE or miR-29b-3p expression was detected by using a Dig-conjugated CRNDE or miR-29b-3p probe on paraffin-embedded colon tissue. Signals were amplified with 3,3′-Diaminobenzidine (DAB), and then the tissues were counterstained with hematoxylin. For the IHC assay, sections were treated with 3% H_2_O_2_/methanol and incubated with an anti-ANGPTL4 antibody (1:1000) at 4 °C overnight after washing with PBS. Sections were allowed to react with horseradish peroxidase polymer-conjugated secondary antibodies, incubated with DAB, and then counterstained with hematoxylin. The staining intensity was scored on a scale of 0~3, as follows: 0 points, negative; 1 point, weakly positive (a low level); 2 points, moderately positive (a moderately high level); and 3 points, strongly positive (a high level).

### 2.17. Statistical Analysis

Results are presented as the mean ± standard deviation (SD). We used Student’s *t*-tests for all comparisons. Statistical analyses of the cell viability and cell migration assays were performed using an unpaired Student’s *t*-test with Excel software. *p* < 0.05 was considered significant.

## 3. Results

### 3.1. CRNDE Is Upregulated in CRC Tissues, and High CRNDE Expression Is Correlated with Poor Prognoses of CRC Patients

Our previous study showed that CRNDE was one of the most significantly upregulated genes in CRC clinical tissues compared to normal colorectal tissues, according to an analysis of a Gene Expression Omnibus (GEO) dataset (GSE21815) (our unpublished data from reference [12]) (Appendix A). We found that the CRNDE level increased about 29-fold in CRC tissues compared to normal colorectal tissues. Next, to understand expression levels of the CRNDE transcript in clinical tissues, we performed an Oncomine [27] analysis to investigate CRNDE transcript levels between tumor and normal tissues in various cancers. As shown in Figure 1A, there were 163 unique analyses of CRNDE. In most of the datasets, CRNDE transcript levels were higher in most tumors compared to normal tissues. The most notable among these tumors was CRC, which showed the greatest number of cases of increased expression levels of the CRNDE transcript. Next, to further confirm expression levels of the CRNDE transcript in a large number of CRC tissues, we analyzed messenger (m)RNA expression profiles of CRNDE transcripts using the GSE21815 dataset and The Cancer Genome Atlas (TCGA) dataset. As shown in Figure 1B,C, significantly increased CRNDE transcripts were found in CRC tissues compared to normal colon tissues. Recently, several papers reported that CRNDE is a crucial tumor promoter. To assess the significance of CRNDE expression in different tumor stages of CRC, we analyzed expression levels of the CRNDE transcript in the GSE21815 and TCGA datasets using CRC tumor samples at different stages. We found that CRNDE exhibited higher expression in a more-advanced stage (IV) than in earlier stages (I/II) (Figure 1D, E). Furthermore, we used the Gene Expression Profiling Interactive Analysis (GEPIA) database [28] to confirm that high CRNDE expression was correlated with a poor OS (Figure 1F) and disease-free survival (Figure 1G) in CRC patients. Collectively, these results indicated that CRNDE was significantly upregulated in CRC patients at advanced tumor-node-metastasis (TNM) stages, and its high expression was correlated with poor outcomes of CRC patients.

### 3.2. CRNDE Promotes Proliferation of CRC Cells

To investigate the functional relevance of CRNDE in CRC cells, we first analyzed CRNDE expression levels in 16 CRC cell lines from the CellExpress database [26] (Figure 2A). Next, high (HCT-116) and low (HCT-15) CRNDE-expressing CRC cell lines were selected to determine the viability and cytotoxicity by manipulating CRNDE expression. Compared to control siRNA-transfected HCT-116 cells, CRNDE siRNA #1 and #2 were able to specifically knock down CRNDE expression by up to 50% (Figure 2B). Knockdown of the endogenous expression of CRNDE in HCT-116 cells caused significant decreases in cell proliferation (Figure 2C, *p* < 0.01 for siRNA #1, *p* < 0.001 for siRNA #2) and colony numbers and sizes compared to control siRNA (Figure 2D, *p* < 0.01 for siRNA #1, *p* < 0.001 for siRNA #2). In contrast, upregulation of CRNDE in GFP-CRNDE-transfected HCT-15 cells (Figure 2E) drastically promoted their growth ability, as shown by increased cell numbers (Figure 2F, *p* < 0.05) and colony numbers (Figure 2G, *p* < 0.05). These results suggest that CRC cell viability and colony numbers significantly decreased following CRNDE-KD but increased in CRNDE-overexpressing CRC cells. Taken together, these findings indicate that CRNDE can markedly promote the proliferation of CRC cells.

### 3.3. Knocking Down CRNDE Inhibited Growth of CRC Cells through Cell Cycle Arrest Not Due to Cell Apoptosis

We then examined whether CRNDE-KD-induced cytotoxicity was mediated by cell cycle effects or apoptotic processes. The knockdown efficiency of CRNDE by CRNDE siRNA #1 and #2 was shown in Appendix A. Experiments were performed using propidium iodide (PI) and Annexin V staining, and antibodies against cell cycle markers and apoptosis markers. Results of the cell cycle distribution revealed that transfection with siCRNDE in HCT-116 cells caused significant accumulation at the G_0_/G_1_ phase (*p* < 0.05 for both CRNDE siRNA #1 and #2) and a decrease in the S phase (*p* < 0.01CRNDE siRNA #2) compared to transfection with control siRNA (Figure 3A,B). Next, HCT-116 cell apoptosis was assessed by Annexin V staining. As shown in Figure 3C,D, transfection with CRNDE siRNA for 48 h produced no significant increase in apoptosis of HCT116 cells compared to control siRNA. According to the above-described results, CRNDE siRNA #2 was used in the following study. Next, cell cycle markers and apoptosis markers were further detected in siCRNDE-transfected cells. The knockdown efficiency of CRNDE by CRNDE siRNA #2 at the concentration of 50 or 100 nM isshown in Appendix A. Results of a Western blot analysis revealed that CRNDE-KD decreased cell proliferation as assessed by induction of p21 expression and inhibition of CDK4 and cyclin D1 expressions (Figure 3E). In addition, transfection with CRNDE siRNA caused the very slight cleavage of caspase-3 and PARP (Figure 3F). However, upregulation of an antiapoptotic protein (Bcl-2) and downregulation of a proapoptotic protein (Bid) were detected in siCRNDE-transfected HCT-116 cells (Figure 3F).Based on the above results, we concluded that CRNDE-KD inhibited proliferation through cell cycle arrest but not by induction of cell apoptosis.

### 3.4. Knocking Down CRNDE Induced Autophagy in CRC Cells

Autophagy is a catabolic process, the activation of which may help cancer cells prevent apoptosis for temporary survival in an adaptation to cellular stress [29]. To determine the effect of CRNDE inhibition on autophagy, we first used a Muse™ Red Fluorescent Protein (RFP)-LC3 Reporter Autophagy Assay Kit, which contained the stably expressing RFP-LC3 Reporter U2OS cell line. Next, control siRNA and siCRNDE were individually transfected into the stably expressing RFP-LC3 Reporter U2OS cell line. As shown in Figure 4A, a shift in the histogram plot was observed in siCRNDE-transfected RFP-LC3 Reporter U2OS cells compared to control siRNA-transfected cells, as indicated by autophagy induction (no autophagy in gray versus induced autophagy in red; Figure 4A, right panel). Statistical results are shown in Figure 4B, which illustrates a significant difference in the mean autophagy intensity of siCRNDE-transfected cells. In addition, induction of autophagy by CRNDE-KD was manifested by increases in the phosphorylation level of adenosine monophosphate-activated protein kinase (AMPK) accompanied by decreases in the phosphorylation level of mammalian target of rapamycin (mTOR) (Figure 4C). LC3 is currently the most widely used autophagosome marker, because the amount of LC3-II reflects the number of autophagosomes and autophagy-related structures. Degradation of p62 is another widely used marker to monitor autophagic activity, because p62 is a polyubiquitin-binding protein known to be degraded during autophagy [30]. Indeed, CRNDE-KD caused increased conversion of LC3-I to LC3-II and a decreased expression level of p62 in HCT-116 cells (Figure 4C). Next, to determine whether autophagy induced by CRNDE-KD could be blocked by the autophagy inhibitor, 3-MA, HCT-116 cells were co-treated with siCRNDE and 3-MA. As shown in Figure 4D, autophagy was induced by CRNDE-KD, and blocked by 3-MA (Figure 4D, lanes 5 and 6). Moreover, to determine whether the effects of CRNDE-regulated cell proliferation are enhanced by modulating autophagy, HCT-116 cells were treated with a combination of siCRNDE and the autophagy inhibitor, chloroquine (CQ), and cell apoptosis was assessed by Annexin V staining. Notably, CQ alone also had a slight inhibitory effect; however, the combination of siCRNDE and CQ led to a significant induction of HCT-116 cell apoptosis (Figure 4E, F), indicating that suppression of CRNDE together with compensatory autophagy caused the demise of cancer cells. Collectively, these results indicated that CRNDE-KD induced autophagy of CRC cells.

### 3.5. CRNDE-KD Inhibits Lipid Metabolism by CRC Cells

Cancer cells tend to activate autophagy via metabolic reprogramming, and autophagy is also a pivotal biological process implicated in metabolic reprogramming, suggesting that metabolic reprogramming and autophagy are often intertwined [31]. Accumulating evidence suggests that activation of AMPK can cause nutrient scarcity by regulating glycolysis or lipid metabolism to promote autophagy [32,33]. Thus, to determine whether CRNDE-KD caused the induction of autophagy through inhibiting glucose or lipid metabolism, we first analyzed glucose uptake. As shown in Figure 5A, glucose uptake was not reduced in CRNDE-KD HCT-116 cells. Next, to measure the glycolytic rate, the extracellular acidification rate (ECAR) was detected. The data indicated that the ECAR was also not decreased in CRNDE-silenced cells (Figure 5B). Next, to determine whether CRNDE-KD caused the inhibition of lipid metabolism, we assessed the inhibitory effect of CRNDE on lipid metabolism by HCT116 cells. BODIPY^505/515^-stained lipophilic bright-green fluorescent dye staining revealed that CRNDE mediated inhibition of about 75% of lipid accumulation in CRNDE-transfected CRC cells compared to control siRNA-transfected HCT116 cells (Figure 5C). The absorbance of BODIPY^505/515^-stained cells was measured and quantified (Figure 5D). Next, to understand alterations of specific lipid metabolism-related genes after CRNDE-KD, the set of genes regulated by CRNDE was retrieved from the GEO dataset (GSE89985). A GSEA revealed that gene sets related to adipogenesis (Figure 5E) were negatively correlated with CRNDE downregulation in CRC cells. Next, to confirm that expressions of adipogenesis-related genes were regulated by CRNDE, an RT-qPCR analysis was performed to measure expressions of the top three adipogenesis-related genes (Figure 5E). Results showed that expressions of the adipogenesis-related genes, myosin light chain kinase (MYLK), glutathione peroxidase 3 (GPX3), and angiopoietin-like 4 (ANGPTL4), were also significantly downregulated in CRNDE-KD cells (Figure 5F). Mechanistically, the absence of ANGPTL4 in hepatocytes was demonstrated to promote AMPK activation. AMPK is activated under low-energy conditions, leading to autophagy [34]. Thus, to investigate potential mechanisms by which loss of ANGPTL4 causes activation of AMPK through controlling lipid metabolism, we analyzed expressions and activities of the main enzymes involved in de novo lipogenesis in HCT116 cells. As shown in Figure 5G, CRNDE-KD led to decreased ANGPTL4 expression, increased phosphorylation of AMPK, increased phosphorylation and consequent inactivation of two AMPK downstream lipid metabolism-associated targets, acetyl-CoA carboxylase (ACC) and hydroxymethylglutaryl-CoA reductase (HMGCR), and reduced the protein expression level of fatty acid synthase (FAS). In addition, activation of ULK1 was detected, which indicated that CRNDE-silenced cells were under lean-energy circumstances. Altogether, these findings suggest that CRNDE plays a critical role in regulating lipid metabolism by CRC cells.

### 3.6. CRNDE Regulates ANGPTL4 Expression via Competitively Binding with miR-29b-3p

A previous study found that ANGPTL4 is highly expressed in CRC [35]. In addition, the roles of ANGPTL4 in glucose and lipid metabolism were recently established in cardiovascular disease [36]. However, the regulatory mechanism of ANGPTL4 involved in energy metabolism by CRC cells remains to be determined. The above-mentioned results demonstrated that CRNDE-KD resulted into the inhibition of ANGPTL4 mRNA and protein expressions by CRC cells. To further investigate whether there was a correlation between CRNDE and ANGPTL4, expression levels of CRNDE and ANGPTL4 in 132 CRC tumor tissues from the GSE21815 database were examined. As shown in Figure 6A, there was a significant positive correlation between expressions of CRNDE and ANGPTL4 in CRC tumor tissues (r = 0.417, *p* < 0.001). LncRNA–miRNA and miRNA–mRNA interactions are generally associated with a variety of biological processes [37]. Accumulating evidence has shown that lncRNAs bind to miRNAs and prevent interactions with their targets; since they prevent miRNAs from completing their regulatory function, lncRNAs acting as sponges are in effect positive regulators of mRNA transcription [38]. It was demonstrated that ANGPTL4 targets binding sites of miR-134-5p [39] and miR-29b-3p [40] according to a reporter assay and RT-qPCR analysis. Thus, we speculated that CRNDE plays a competitive role as endogenous RNA (ceRNA) by sponging miR-134-5p or miR-29b-3p to regulate ANGPTL4 protein expression. To test this hypothesis, we first determined the effects of CRNDE on miR-134-5p or miR-29b-3p expressions. As shown in Figure 6C, CRNDE-KD resulted in an obvious increase in the expression of miR-29b-3p, but not in the expression of miR-134-5p (Figure 6B) in HCT-116 cells. Further, to determine whether CRNDE participates in regulating miR-29b-3p expression, we investigated expressions of CRNDE and miR-29b-3p in paired CRC resected tumor tissues and corresponding adjacent non-tumor tissues obtained from a public GEO dataset (GSE32323). As shown in Figure 6D, we observed that the CRNDE transcript was significantly upregulated in tumor tissues (*p* < 0.001). Inversely, miR-29b-3p expression was significantly decreased in CRC tumor tissues compared to corresponding adjacent non-tumor tissues (Figure 6E). A correlation analysis also showed a negative correlation between CRNDE and miR-29b-3p expression levels in 34 CRC resected tumors and corresponding adjacent non-tumor tissues (r = −0.504, *p* < 0.01, Figure 6F). To further probe the direct relationship between CRNDE and miR-29b-3p, we constructed dual luciferase reporters of CRNDE, which contained the potential miR-29b-3p-binding site through an miRTarBase database analysis [41] and the mutant miR-29b-3p-binding site of CRNDE (Figure 6G). Results showed that miR-29b-3p mimics significantly reduced luciferase activity of the WT CRNDE reporter compared to the negative control, while miR-29b-3p mimics posed no impact on the luciferase activity of the CRNDE mutant-type reporter (Figure 6H). The above results demonstrated that CRNDE can regulate ANGPTL4 expression via competitive binding to miR-29b-3p.

### 3.7. High Levels of CRNDE and ANGPTL4 and ALow Level of miR-29b-3p in CRC Tissues Are Involved in Regulating Lipid Metabolism by the miR-29b-3p/ANGPTL4 Axis-Mediated Regulation of AMPK/ULK1signaling

Next, we enrolled three serial sections of a colon adenocarcinoma tissue array (BioMax, Rockville, MD, USA) to evaluate the prognostic values of CRNDE, miR-29b-3p, and ANGPTL4 in CRC tissues and found that CRC tumors expressed high CRNDE and ANGPTL4 levels but a low miR-29b-3p level (Figure 7A). Among 50 cases of CRC tissues, high levels of CRNDE and ANGPTL4 were found in about 80% of CRC tumors (Figure 7B). To investigate whether the phenotype of miR-29b-3p overexpression is similar to CRNDE-KD, we first transfected the HCT-116 cell line with an miR-29b-3p mimic with relative low expression of miR-29b-3p [42]. Compared to transfection with the negative control, results showed that transfection with the miR-29b-3p mimic resulted in about a 104-fold increase in mature miR-29b-3p in the HCT-116 cell line examined at a time course of 48 h (Figure 7C). Next, to determine whether miR-29b-3p overexpression caused the inhibition of lipid metabolism, we assessed the inhibitory effect of miR-29b-3p on lipid metabolism in HCT-116 cells. BODIPY^505/515^ staining with the lipophilic bright-green fluorescent dye revealed that miR-29b-3p mediated about 75% inhibition of lipid accumulation in miR-29b-3p-transfected CRC cells compared to control miRNA-transfected HCT-116 cells (Figure 7D,E). As expected, there was a significant reduction in the ANGPTL4 protein amount and increases in phosphorylation levels of AMPK and ULK1, accompanied by the consequent inactivation of ACC and HMGCR, as well as a reduced protein expression level of FAS in miR-29b-3p mimic-transfected HCT-116 cells (Figure 7F). Taken together, these findings proved that CRNDE silencing induced autophagy of CRC cells by the miR-29b-3p-regulated inhibition of ANGPTL4, which caused inhibition of de novo lipogenesis (Figure 7G).

## 4. Discussion

Accumulating evidence supports that lncRNAs play major roles in human physiological and pathophysiological processes [43]. LncRNA CRDNE acts as an oncogene in many human cancers [44,45,46]. However, little is known about the roles and biological mechanisms of CRNDE in the physiological effects of CRC. In this study, we demonstrated that loss of CRNDE triggered autophagy through regulating metabolic signaling. Herein, we summarize the evidence that supports this conclusion. First, we demonstrated that CRNDE-KD inhibited proliferation through cell cycle arrest but not induction of cell apoptosis. Second, we found that CRNDE-KD caused induction of autophagy of CRC cells. Third, we found that CRNDE plays critical roles in regulating glucose and lipid metabolism of CRC cells via competitively binding miR-29b-3p to regulate ANGPTL4 expression. Fourth, we found that CRNDE-KD caused induction of autophagy of CRC cells through miR-29b-3p-regulated inhibition of ANGPTL4, thereby inhibiting lipogenesis. Collectively, such a conclusion could be drawn that knockingdown CRNDE prevented the malignant behaviors and induced autophagy of CRC cells, thereby reducing lipid accumulation in CRC cells through the miR-29b-3p/ANGPTL4 axis.

Various oncogenic pathways may contribute to CRC carcinogenesis [47]; however, the potential involvement of lncRNA(s) in physiological regulation of autophagy by metabolic circuitries is poorly defined in human CRC. Metabolic stress often occurs in solid tumors, which includes rapidly proliferating tumor cells that lack sufficient nutrient and oxygen supplies [48]. To overcome this metabolic hurdle, tumor cells engage in autophagy and metabolic alterations to increase intracellular nutrient supplies. Hence, autophagy plays a prosurvival role in tumor development [49]. In this study, we found that CRNDE-KD could induce autophagy, which was confirmed by evaluating expressions of autophagy markers and by a flow cytometric analysis. Meantime, we used the autophagy inhibitor, CQ, to investigate the function of autophagy in CRNDE-KD. Inhibition of autophagy reduced CRC cell growth. The autophagy pathway and metabolism signaling closely communicate with each other, and this is regulated by the AMPK/mTOR pathway [34]. AMPK plays a role in autophagy induction under lean-energy circumstances by phosphorylating the mTOR component, Raptor, leading to inactivation of mTOR and subsequent activation of the ULK1 complex [50]. In addition, AMPK was reported to play a key role in controlling overall cellular lipid metabolism [51]. In this study, we found that CRNDE-KD led to increased phosphorylation and consequent inactivation of two AMPK downstream lipid metabolism-associated targets, ACC and HMGCR, as well as reducing the FAS protein expression level. In brief, our results supported that CRNDE-KD attenuated lipid accumulation and improved lipid metabolism in CRC cells, and AMPK and mTOR are the main signaling integrators and modulators of autophagy and lipid metabolism.

Numerous studies expounded that miRNAs participate in tumorigenesis and that mRNA expressions can be directly regulated by miRNAs [37]. Previous studies showed that miR-29b-3p acts as a tumor suppressor in various cancers [42,52,53,54,55], and it was shown to restrain multiple oncogenic processes, including by promoting tumor cell apoptosis, by suppressing DNA methylation of tumor-suppressor genes, by reducing tumor proliferation, and by increasing chemo-sensitivity [56]. Although miR-29b-3p has been thoroughly documented as a tumor suppressor in regulating multiple oncogenic processes, the role of miR-29b-3p-mediated regulation of cancer metabolism is still unclear. In this study, we demonstrated that miR-29b-3p-regulated inhibition of ANGPTL4 caused inhibition of lipid metabolism. ANGPTL4 is associated with a poor prognosis of patients with various solid tumors, suggesting an important role in cancer onset and progression [57]. ANGPTL4 is best known for its role as an adipokine involved in regulating lipid metabolism [58]. Although ANGPTL4 was demonstrated to be the direct target of miR-29b-3p in osteosarcomas [40], the regulatory mechanism of ANGPTL4 in lipid metabolism of CRC cells remains unclear. Additionally, numerous CRC-associated lncRNA/miRNA/mRNA axes have been reported in recent studies; they are mostly involved in CRC cell proliferation, migration, invasion, tumor growth, and metastasis [59], but rarely related to CRC energy metabolism. In this study, we found that CRNDE could directly bind to miR-29b-3p, which could prevent miR-29b-3p-mediated inhibition of ANGPTL4 expression in CRC cells. Thus, knocking down CRNDE can reduce lipid accumulation through the miR-29b-3p/ANGPTL4 axis and consequently induce autophagy of CRC cells.

In summary, our current study demonstrated that CRNDE and ANGPTL4 are upregulated, while miR-29b-3p is downregulated in CRC tumor tissues. We showed that silencing of CRNDE reduced lipid accumulation and induced autophagy of CRC cells. This is the first study to discover and prove that CRNDE can competitively bind miR-29b-3p, and described a novel CRNDE/miR-29b-3p/ANGPTL4 signaling pathway with a regulatory function in CRC. The findings show that CRNDE plays an important role in CRC, and the present study provides evidence of crosstalk among CRNDE, miR-29b-3p, and ANGPTL4, thereby shedding new light on potential therapeutic targets for CRC treatment.

## 5. Conclusions

CRNDE is significantly upregulated in CRC patients, and its high expression is related to poorer prognoses of CRC patients. Knockdown of CRNDE caused the induction of autophagy of CRC cells, and suppression of CRNDE together with compensatory autophagy caused the demise of cancer cells. In addition, we found that CRNDE plays a critical role in regulating lipid metabolism of CRC cells via competitively binding with miR-29b-3p to regulate ANGPTL4 expression. Moreover, our results supported that knockingdown CRNDE attenuated lipid accumulation and improved lipid metabolism in CRC cells, and AMPK and mTOR are the main signaling integrators and modulators of autophagy and lipid metabolism. Collectively, these findings provide evidence of the crosstalk among CRNDE, miR-29b-3p, and ANGPTL4, shedding new light on potential therapeutic targets for CRC treatment.

## Figures and Tables

**Figure 1 biomedicines-09-01438-f001:**
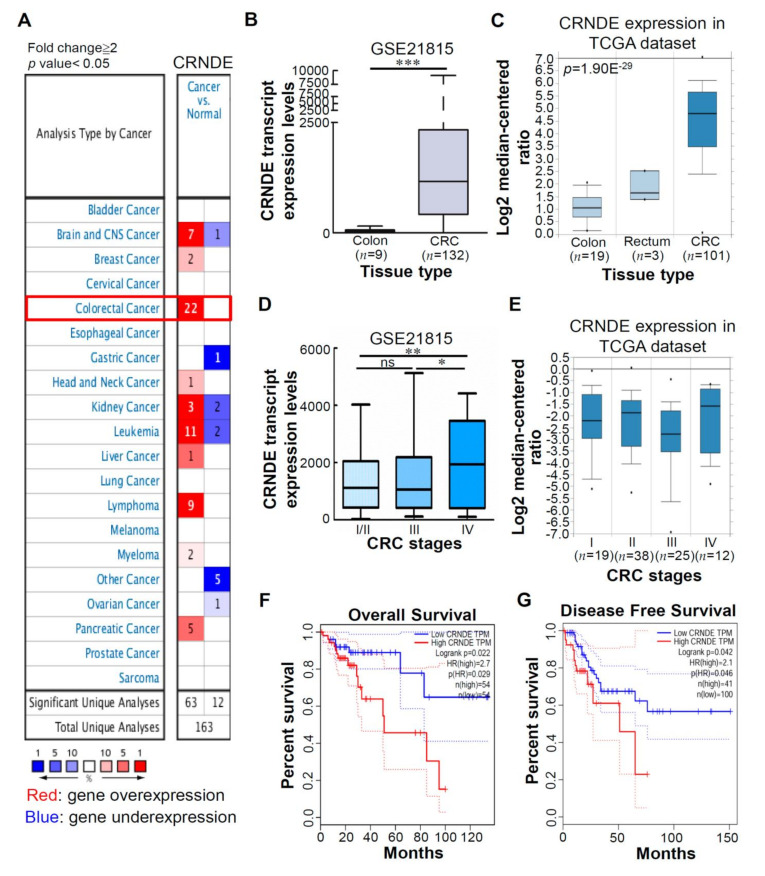
Relative expression of colorectal neoplasia differentially expressed (CRNDE) in colorectal cancer (CRC) tissues.(**A**) Expressions of CRNDE mRNA in 20 common cancers were compared with those in corresponding normal tissues (in the Oncomine Database). The search criteria thresholds for datasets of cancer versus normal analysis were a multiple of change of ≥2, a *p* value of <0.05, and a gene rank in the top 10%. Red represents gene overexpression in the analyses; blue represents gene under-expression. (**B**) Relative CRNDE expression in human CRC tissues compared to noncancerous tissues via a GSE21815 data analysis. (**C**) Relative expression levels of CRNDE in normal colon/rectum tissues and CRC tissues using the TCGA database. (**D** and **E**) Data are presented as relative expression levels in tumor tissues. CRNDE expression was significantly increased in patients at a higher pathological stage and with larger tumors. Kaplan–Meier analysis of overall survival (**F**) and disease-free survival (**G**) of CRC patients with the corresponding expression profiles: CRNDE (low) and CRNDE (high). Log-rank analysis was used for comparison between groups.* *p* < 0.05, ** *p* < 0.01, *** *p* < 0.001. ns: non-significance.

**Figure 2 biomedicines-09-01438-f002:**
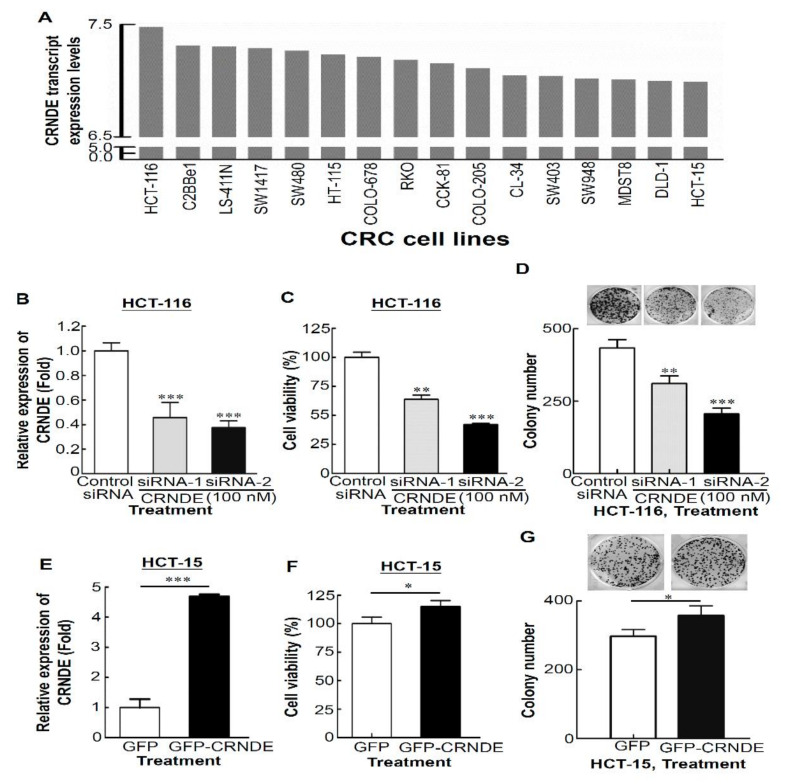
Colorectal neoplasia differentially expressed (CRNDE) regulates the proliferation of colorectal cancer (CRC) cells. (**A**) Expression levels of CRNDE in 16 CRC cell lines were obtained from the *CellExpress* database. (**B**) CRNDE levels in HCT-116 cells after siRNA-mediated knockdown of CRNDE were detected by an RT-qPCR. (**C**) An MTT assay was performed to determine the proliferation of CRNDE-depleted HCT-116 cells. (**D**) A colony-forming assay was performed to determine the effects of CRNDE depletion on the growth of HCT-116 cells. (**E**) Expression levels of CRNDE in green fluorescent protein (GFP)-CRNDE-transfected HCT-15 cells. The GFP-CRNDE-regulated cell proliferation of HCT-15 cells by an MTT assay analysis (**F**) and colony-forming assay (**G**).* *p* < 0.05, ** *p* < 0.01, *** *p* < 0.001.

**Figure 3 biomedicines-09-01438-f003:**
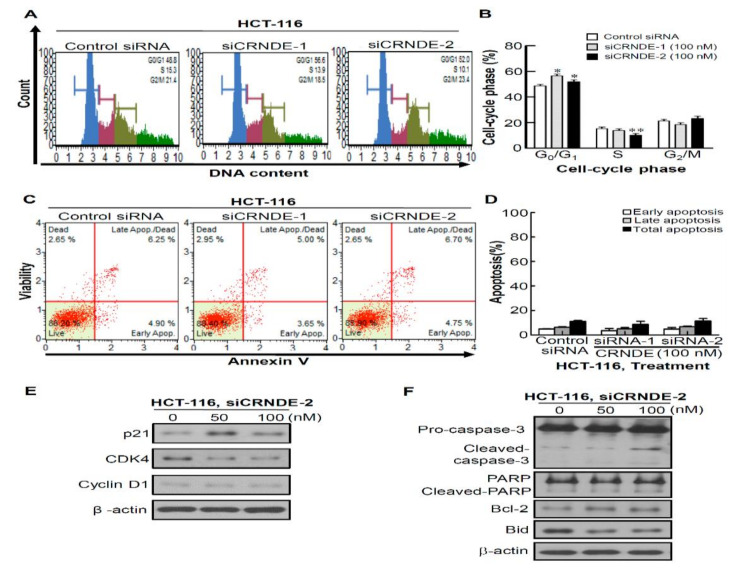
Functional roles of colorectal neoplasia differentially expressed (CRNDE) in regulating colorectal cancer (CRC) cell growth. (**A**) HCT-116 cells were stained with propidium iodide (PI) and analyzed using a Muse™ Cell Analyzer. (**B**) The quantification result of PI-positive cells with CRNDE-knockdown. (**C**) HCT-116 cells were stained with Annexin V-FITC and analyzed using a Muse™ Cell Analyzer. (**D**) Quantification of results of Annexin V-positive cells with CRNDE-knockdown. Knockdown of CRNDE-induced cytotoxicity is mediated by cell cycle regulators (**E**) or apoptotic regulators (**F**). Actin was used as a loading control. * *p* < 0.05, ** *p* < 0.01.

**Figure 4 biomedicines-09-01438-f004:**
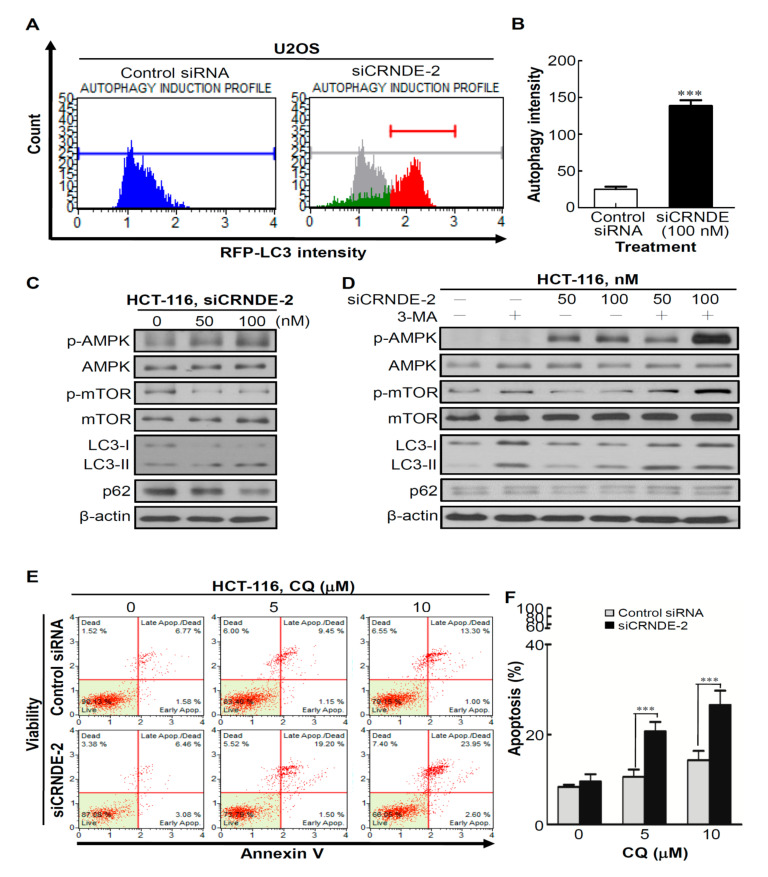
Colorectal neoplasia differentially expressed (CRNDE) is involved in regulating autophagy. (**A**) Small interfering (si)CRNDE-transfected red fluorescent protein (RFP)-LC3 reporter U2OS cells showed a shift in the histogram plot indicating autophagy induction (no autophagy in gray versus induced autophagy in red). (**B**) Quantification results of the autophagy intensity. (**C**) Western blot analysis of the effects of CRNDE-knockdown on adenosine monophosphate-activated protein kinase (AMPK) phosphorylation and its downstream targets, mammalian target of rapamycin (mTOR) and autophagy markers in HCT-116 cells. (**D**) Whole-cell lysates were prepared from HCT-116 cells co-treated with siCRNDE and 3-MA and immunoblotted with antibodies against AMPK, mTOR, and p62. (**E**) CRNDE was silenced for 48 h with 100 nM of specific siRNA, following which cells were treated with chloroquine (CQ; an autophagy inhibitor) at the indicated concentrations for an additional 48 h. Caspase-3/7 activities were measured with a Muse Cell Analyzer. (**F**) Quantification results of Annexin V-positive cells with CRNDE-knockdown alone or in the presence of CQ. *** *p* < 0.001.

**Figure 5 biomedicines-09-01438-f005:**
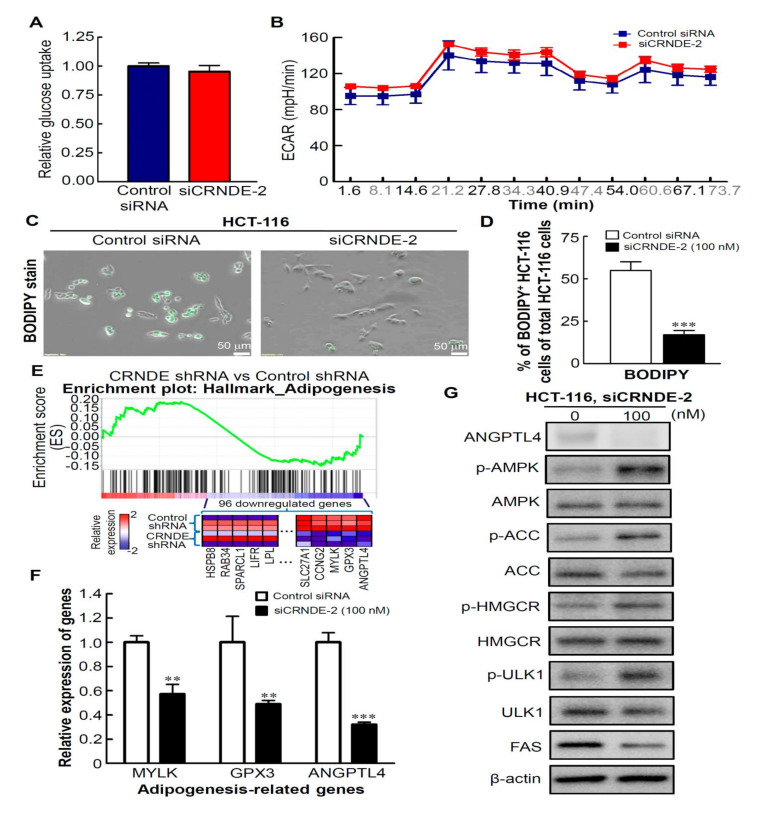
Knockdown of colorectal neoplasia differentially expressed (CRNDE) caused inhibition of lipid metabolism of colorectal cancer (CRC) cells. (**A**) Glucose consumption was determined in CRNDE-knockdown HCT-116 cells. (**B**) Analysis of the extracellular acidification rate (ECAR) of HCT-116 cells with CRNDE-knockdown. (**C**) HCT-116 cells were transfected with CRNDE small interfering (si)RNA for 48 h. Images of BODIPY^505/515^-stained cells were captured with a fluorescence microscope. (**D**) BODIPY^505/515^-stained results were quantified and are represented as multiples of change, considering the control siRNA cell value as 1-fold. Error bars represent the mean ± standard deviation (SD). (**E**) A gene set enrichment analysis (GSEA) showed enrichment of downregulated adipogenesis gene sets in stable CRNDE-knockdown CRC cells. Heat-map image illustrating gene expression levels of the leading edge subset. (**F**) Expressions of representative genes (*MYLK*, *GPX3*, and *ANGPTL4*) in CRNDE-knockdown HCT-116 cells. (**G**) Western blot analysis of the effects of CRNDE-knockdown on the phosphorylation and expression levels of lipid metabolism-associated targets in HCT-116 cells, including the phosphorylation levels of acetyl-CoA carboxylase (ACC) and hydroxymethylglutaryl-CoA reductase (HMGCR), as well as fatty acid synthase (FAS) protein level. ** *p* < 0.01, *** *p* < 0.001.

**Figure 6 biomedicines-09-01438-f006:**
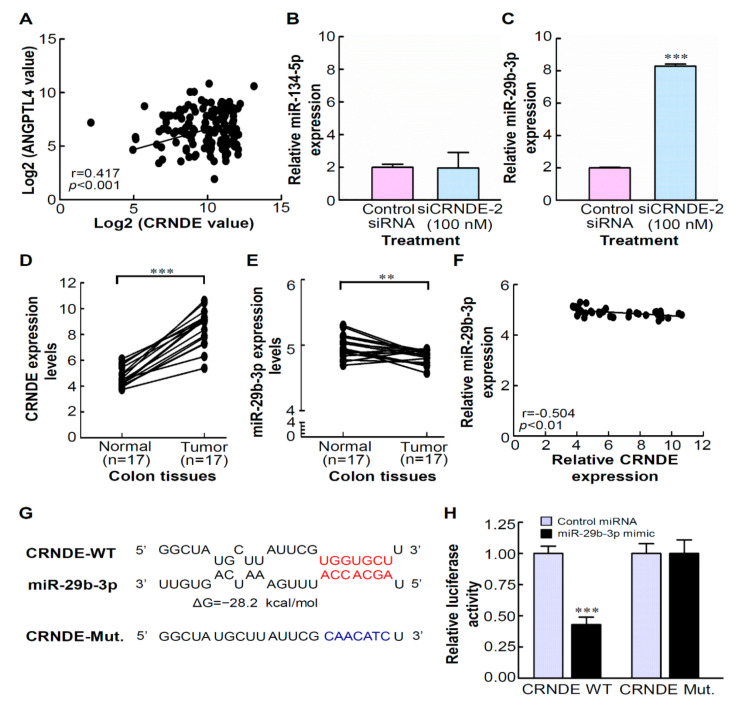
Colorectal neoplasia differentially expressed (CRNDE) directly interacts with miR-29b-3p. (**A**) Correlation analysis revealed the positive relationship between CRNDE and angiopoietin-like 4 (ANGPTL4) expressions in 132 colorectal cancer (CRC) tumor tissues. MiR-134-5p (**B**) and miR-29b-3p (**C**) expressions were determined by an RT-qPCR in CRNDE-knockdown HCT-116 cells. Expressions of CRNDE (**D**) and miR-29b-3p (**E**) in 17 normal/tumor (NT) pairs of CRC resected tumor (T) tissues and corresponding adjacent non-tumor (N) tissues obtained from a public GEO dataset (GSE32323). (**F**) Correlation analysis revealed a negative relationship between CRNDE and miR-29b-3p expressions in 34 cases of NT pairs of CRC tissues from the GEO dataset (GSE32323). (**G**) A bioinformatics analysis revealed predicted binding sites between CRNDE and miR-29b-3p. (**H**) A luciferase reporter assay demonstrated miR-29b-3p mimics significantly decreased the luciferase activity of CRNDE-wild type (WT) in HCT-116 cells, while miR-29b-3p mimics did not affect the luciferase activity of CRNDE-mutant (Mut). ** *p* < 0.01, *** *p* < 0.001.

**Figure 7 biomedicines-09-01438-f007:**
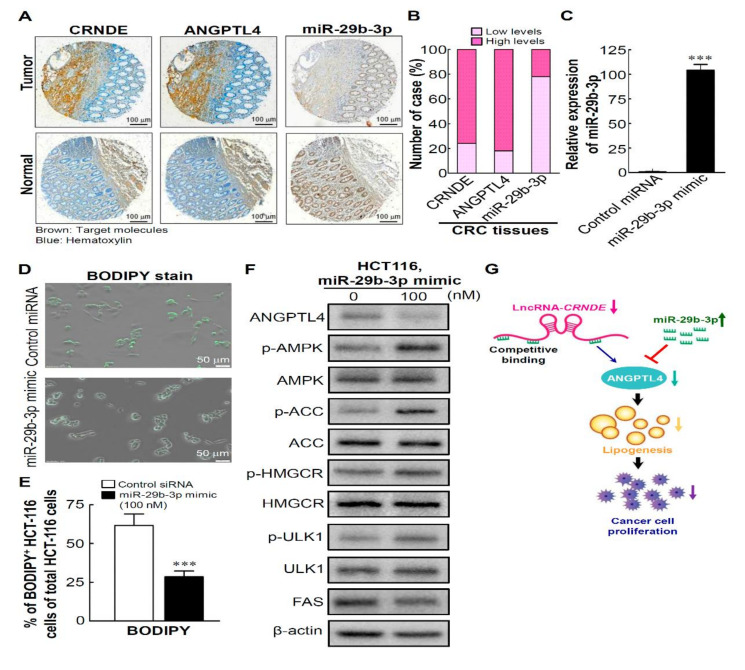
High levels of colorectal neoplasia differentially expressed (CRNDE) and angiopoietin-like 4 (ANGPTL4) and a low level of miR-29b-3p in colorectal cancer (CRC) tissues involved in regulating lipid metabolism. (**A**) Levels of CRNDE, miR-29b-3p, and ANGPTL4 in CRC tissues were examined by IHC and ISH. (**B**) Percentage of cases are plotted on the y-axis, and the type of molecule is plotted on the x-axis. (**C**) Validation of miR-29b-3p expression levels after transfection with a miR-29b-3p mimic for 48 h in the HCT-116 cell line. (**D**) HCT-116 cells were transfected with a miR-29b-3p mimic for 48 h. Images of BODIPY^505/515^-stained cells were captured with a fluorescence microscope. (**E**) BODIPY^505/515^-stained results were quantified and are presented as multiples of change, considering the control miRNA cell value as 1-fold. Error bars represent the mean ± standard deviation (SD). (**F**) Western blot analysis of the effects of miR-29b-3p overexpression on the phosphorylation and expression levels of lipid metabolism-associated targets in HCT-116 cells. (**G**) Schematic model showing that CRNDE silencing induced autophagy of CRC cells by the miR-29b-3p-regulated inhibition of ANGPTL4, causing the inhibition of de novo lipogenesis. *** *p* < 0.001.

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
