# Peer review of "A New Light on Potential Therapeutic Targets for Colorectal Cancer Treatment"

_biomedicines, 2021, doi:10.3390/biomedicines9101438_

Round 1

Reviewer 1 Report

Major point;

Fig. 1; The CRNDE expression is not different between Stages I, II, and III, and higher only in stage IV. Therefore, the survival curves (F and G) only show the well-known fact that the prognosis of Stage IV CRC cases is worse than that of I / II / III.

Author Response

We appreciate the comments of this reviewer and believe that our manuscript has been improved by attention to him or her. Our responses to the specific issues raised by this reviewer are shown in attachment.

Reviewer 2 Report

1.PLEASE CHANGE THE TITLIE in:

A new light on potential therapeutic targets for CRC treatment.

2."Our findings show that CRNDE plays an important role in CRC, and the present study provides evidence of crosstalk among CRNDE, miR-29b-3p, and ANGPTL4, thereby shedding new light on potential therapeutic targets for CRC treatment." - PLEASE DELETE THIS!

3."CRC cell lines were provided by Prof. YW Cheng and Prof. H Lee (Graduate Institute of Cancer Biology and Drug Discovery, Taipei Medical University). - PLEASE REPLECE WITH - "CRC cell lines were provided by Graduate Institute of Cancer Biology and Drug Discovery, Taipei Medical University.

4."Collectively, these findings show that CRNDE plays an important role in CRC, and the present study provides evidence of the crosstalk among CRNDE, miR-29b-3p, and ANGPTL4, shedding new light on potential therapeutic targets for CRC treatment" - PLEASE REPLACE WITH ""Collectively, these findings provides evidence of the crosstalk among CRNDE, miR-29b-3p, and ANGPTL4, shedding new light on potential therapeutic targets for CRC treatment"

Author Response

(The authors gave the same response as above.)

Reviewer 3 Report

In this paper, the authors study the effect of Lnc RNA CRNDE on lipid metabolism that induces autophagy by the miR-29b-3p/ANGPTL4 axis in CRC cells. Overall, the explanations are strong and well written. However, the manuscript raises several questions to be addressed.

Major comments:

  1. In Figure 1, the authors show the TCGA dataset from Oncomine. I know that Oncomine shows various datasets. Authors need to show CRNDE data from Oncomine's various datasets. In Figure 1A, the authors describe 163 datasets for CRNDE. However, there are 22 colon cancer datasets. Therefore, the author should clarify the results or explanations.
  2. In Figure 2, the authors describe the effect of CRNDE on the proliferation of CRC cells. The authors should show the growth curves of CRCs to clarify the colony formation data.
  3. In Figure 3C, Annexin V staining was performed to account for apoptosis. The author should explain what viability in Y-axis is. Does viability mean PI staining?
  4. In Figure 3, the authors used siRNA against CRNDE. qPCR data are required to clarify CRNDE inhibition.
  5. In Figure 3E and 3F, 50 nM of siCRNDE-2 increased the expression of p21 but not induced cleaved caspase 3, and 100 nM showed no increase in p21 expression but increased cleaved caspase 3. With these results, the authors need to clarify the signaling pathway.
  6. In Figures 4A and 4B the authors used USO2 cells, not colorectal cancer cells. Is CRNDE overexpressed in USO2 cells? Authors should clarify data. In Figure 4D, 3-MA was used as an autophagy inhibitor, and in Figures 4E and 4F, chloroquine was used. Authors should explain why other inhibitors were used in the experiment.
  7. In Figures 4C and 4D, the p62 signal from the western blot has a different pattern. In Figure 4C, the signal is one lane, but in Figure 4D, the signal is two. Authors should clarify data.
  8. In lanes 5 and 6 of Figure 4D, phosphorylation of AMPK did not inhibit phosphorylation of mTOR. In addition, 3-MA, an autophagy inhibitor, did not inhibit LC3-I and II. Authors should clarify data.

Author Response

(The authors gave the same response as above.)

Round 2

Reviewer 1 Report

My question was not adequately answered. I agree that it is important to investigate the relationship between tumor malignancy and CRNDE level. But authors can not conclude the positive relationship between them in clinical setting in Fig 1 ( no data about clinicopathological background, nor multivariate  analysis and no difference between stage I, II and III).

Author Response

We appreciate the comments of this reviewer and believe that our manuscript has been improved by attention to him or her. Our responses to the specific issues raised by this reviewer is shown in attachment.

Reviewer 3 Report

This reviewer appreciates the extra effort of the authors to address the concerns raised. The authors have properly addressed most of the requirements and suggestions. 

Author Response

Dear Reviewer,

Thank you for your kind comments.

Best Regards,

Kuen-Haur Lee

Round 3

Reviewer 1 Report

This time, questions are well answered. This paper is thought to be ready for publish.